# Synergistic Effect of Perampanel and Temozolomide in Human Glioma Cell Lines

**DOI:** 10.3390/jpm11050390

**Published:** 2021-05-10

**Authors:** Andrea Salmaggi, Cristina Corno, Marta Maschio, Sara Donzelli, Annachiara D’Urso, Paola Perego, Emilio Ciusani

**Affiliations:** 1Stroke Unit-Department of Neurology, Manzoni Hospital,23900 Lecco, Italy; a.salmaggi@asst-lecco.it; 2Department of Diagnostic and Technology, Fondazione IRCCS Istituto Neurologico Carlo Besta, 20133 Milan, Italy; cristina.corno@istitutotumori.mi.it (C.C.); annachiara.durso@istituto-besta.it (A.D.); 3Molecular Pharmacology Unit, Department of Applied Research and Technological Development, Fondazione IRCCS Istituto Nazionale dei Tumori, 20133 Milan, Italy; paola.perego@istitutotumori.mi.it; 4Center for Tumor-Related Epilepsy, UOSD Neuroncology Regina Elena National Cancer Institute Rome, 00161 Rome, Italy; marta.maschio@ifo.gov.it; 5Oncogenomic and Epigenetic Unit, Regina Elena National Cancer Institute, 00161 Rome, Italy; sara.donzelli@ifo.gov.it

**Keywords:** glioblastoma, perampanel, AMPA receptors, glutamate

## Abstract

Glioblastoma is characterized by a high proliferative rate and drug resistance. The standard of care includes maximal safe surgery, followed by radiotherapy and temozolomide chemotherapy. The expression of glutamate receptors has been previously reported in human glioma cell lines. The aim of this study was to examine the cellular effects of perampanel, a broad-spectrum antiepileptic drug acting as an α-amino-3-hydroxy-5-methyl-4-isoxazolepropionic acid receptor (AMPA) glutamate receptor antagonist, alone or in combination with temozolomide. Four human glioma cell lines were exposed to different concentrations of perampanel and temozolomide, alone or in combination. The type of drug interaction was assessed using the Chou-Talalay method. Apoptosis, cell cycle perturbation, and glutamate receptors (GluRs) subunit expression were assessed by flow cytometry. Perampanel significantly inhibited the growth, inducing high levels of apoptosis. A strong synergistic effect of the combination of perampanel with temozolomide was detected in U87 and A172, but not in U138. Treatment with perampanel resulted in an increased GluR2/3 subunit expression in U87 and U138. Perampanel displays a pro-apoptotic effect on human glioblastoma cell lines when used alone, possibly due to increased GluR2/3 expression. The observed synergistic effect of the combination of temozolomide with perampanel suggests further investigation on the impact of this combination on oncologic outcomes in glioblastoma.

## 1. Introduction

Gliomas are the most frequent malignant primary intracranial tumors, with an estimated incidence of 6–8 cases/100.000/year. With the exception of grade I glioma, these tumors invariably progress intracranially, despite all available treatment modalities, leading to disability and death. Life expectancy depends on tumor grade and molecular features, ranging from a median of 14.6 months in glioblastoma (GBM) after surgery, radiation therapy, and temozolomide chemotherapy to a median of more than 15 years in grade II oligodendroglioma with a favorable molecular profile (i.e., 1p/19q co-deletion) and clinical features (age, neurological conditions, tumor size) [1,2].

Mutations at multiple levels lead to deranged cell function, uncontrolled proliferation, and migration of glioma cells [3]. A number of pathways and mediators are involved in these features; among these, the glutamatergic pathway has been shown to be upregulated in high-grade glioma with enhanced release of glutamate by high grade glioma cells, which sustain glioma cell proliferation and migration [4].

Glutamate is produced in excess by tumor cells, induces neuronal cell death, and enhances proliferation and resistance to apoptosis, as well as migration of tumor cells [5]. Increased expression of glutamate receptors has also been detected in GBM-derived brain tumor-initiating cells [6]. Of note, glutamate is not only implicated in the growth of glioma, but also in induction of seizures, a frequent clinical manifestation found in glioma [7].

Glutamate receptors are classified in three subfamilies: two ligand-gated ionotropic receptors (the N-methyl-D-aspartate or NMDA receptors and the α-amino-3-hydroxy-5-methyl-4-isoxazolepropionic acid or AMPA receptors) and one metabotropic glutamate receptor (mGluR). Previously, various metabotropic glutamate receptor and AMPA receptor antagonists, such as JNJ165968, a mGluR1 antagonist, and LY341495, a mGluR2/3 antagonist, were described to display anticancer activity [8].

Inhibitors of the glutamatergic pathway have been investigated in this scenario; a clinical phase II talampanel trial in GBM [9], although with promising results, has not led to phase III clinical trials aiming at prolonging survival of GBM patients, while research in epilepsy has led to the approval of perampanel as an add-on treatment in partial seizures (with or without secondary generalization) and as add-on treatment for primary generalized tonic-clonic seizures in patients older than 12 years with idiopathic generalized epilepsy [10]. Both talampanel and perampanel are selective non-competitive AMPA receptor antagonists. However, compared to talampanel, perampanel has a 5-fold longer half-life in humans and excellent blood-brain barrier penetration [11,12]. Thus, use of perampanel in the management of glioma might offer an advantage, not only in control/prevention of seizures, but also in controlling tumor growth.

However, the cellular effects of perampanel on glioma cell lines have been poorly investigated.

In this pre-clinical investigation, we aimed at assessing the impact of perampanel alone or in combination with temozolomide on the growth of human high-grade glioma cells, including glioblastoma and a grade III astrocytoma cell lines. The effect of treatment with perampanel on expression of AMPA receptors subunits (GluR1–4) was also evaluated, despite the pro-apototic effects of the drug.

## 2. Materials and Methods

### 2.1. Cell Lines and Drugs

The human glioblastoma cell lines U87 (#HTB-14), U138 (#HTB-16), and A172 (#CRL-1620) and the grade III astrocytoma cell line SW1783 (#HTB-13) were from ATCC. All glioma cell lines were grown in DMEM medium (Lonza, Basel, Switzerland), supplemented with 10% FBS (Gibco, Life Technologies, Carlsbad, CA, SUA). Cells were routinely checked for mycoplasma contamination (Lonza, Basel, Switzerland), used within 20 passages from thawing from a frozen stock. For in vitro studies, perampanel (Eisai Co., Ibaraki, Japan) and temozolomide (Schering Plough Corporation, Kenilworth, NJ, USA) were dissolved in dimethylsulfoxide (DMSO, Sigma-Aldrich, St. Louis, MI, USA). Final DMSO concentration in medium never exceeded 0.25%.

### 2.2. Cell Growth Inhibition and Drug Interaction Analyses

Cells were seeded in 12-well plates and, 24 h later, exposed to increasing concentrations of perampanel, temozolomide, or a combination. For combination studies, the cells were pre-incubated with perampanel for 1 h prior to treatment, with temozolomide for 72 h. At the end of treatment, cells were harvested using trypsin and cell growth inhibition was evaluated by counting cells with an automatic instrument (Z2 Particle Counter, Beckman Coulter, Milan, Italy). All experiments were performed at least three times. IC_50_ is defined as the concentration of the drug inhibiting cell growth by 50%. Drug interaction was evaluated according to the Chou–Talalay method, assigning a combination index (CI) value to each drug combination using the Calcusyn software (Biosoft, Cambridge, United Kingdom) [13]. CI values lower than 0.85–0.90 indicate synergistic drug interactions, whereas CI values higher than 1.20–1.45 or around 1 stand for antagonism or an additive effect, respectively. A moderate synergism can be defined for CI values between 0.85 and 0.7, whereas a marked synergism is evident for CI values lower than 0.5.

### 2.3. Analysis of Apoptosis

Apoptosis was evaluated by the Annexin V-binding assay (Immunostep, Salamanca, Spain) in glioma cells treated for 24 h with perampanel and harvested at the end of treatment or 24 h later. Cells were resuspended in binding buffer (10 mM HEPES-NaOH, pH 7.4, 2.5 mM CaCl2, and 140 mM NaCl, Immunostep, Salamanca, Spain). A fraction of 10^5^ cells were incubated in binding buffer at room temperature in the dark for 15 min with 5 µL of FITC-conjugated Annexin V and 10 μL of 2.5 μg/mL propidium iodide (Immunostep, Salamanca, Spain). Annexin V binding and propidium iodide (PI) staining were detected by flow cytometry (BD Accuri C6, Becton Dickinson, Milan, Italy). At least 10^4^ events/sample were acquired and analyzed using the instrument software.

### 2.4. Cell Cycle Analysis

Glioma cells were treated for 24 h with perampanel, and cell cycle perturbations were measured immediately after the end of treatment or 24 h later, using flow cytometry (BD Accuri C6). Floating and adherent cells were harvested, washed with saline, then fixed in 70% cold ethanol and incubated overnight at 4 °C with PBS containing 50 μg/mL PI (Sigma-Aldrich, Milan, Italy) and 1 mg/mL RNase A (Sigma-Aldrich, Milan, Italy). At least 2 × 10^5^ cells were collected and evaluated for DNA content. Cell cycle distribution was analyzed using FlowJo 10 (Becton Dickinson).

### 2.5. Glutamate Receptor Subunit Membrane Expression

AMPA receptor subunit membrane expression was evaluated by flow cytometry. Briefly, cultured cells were harvested, washed once in complete medium, and counted. In total, 150,000 cells/250 µL were incubated for 1 h at 4 °C, with 1 µg of the following primary antibodies: rabbit anti-GluR1, rabbit anti-GluR2&3, and rabbit anti-GluR4 (Chemicon, Temecula, CA, USA). Cells were then washed once in complete medium and incubated with 5 µL of FITC-conjugated goat anti rabbit IgG (#6717, Abcam, Milan, Italy) for 1 h at 4 °C. GluRs expression was detected by flow cytometry (Navios EX, Bekman Coulter). At least 10^4^ events/sample were acquired and analyzed using a specific software (Navios EX Softwar, Bekman Coulter). Data are expressed as the ratio between the mean fluorescence intensity (MFI) of the specific antibody and the MFI of the relative isotypic control (Simultest, γ1/γ2a, Becton Dikinson, Franklin Lakes, NJ, USA). MFI ratio values greater than 1 indicate expression of the molecule on the cell surface.

### 2.6. Western Blot Analysis

Western blot analysis was carried out as described [14]. Briefly, samples were fractionated by SDS-PAGE and blotted on nitrocellulose membranes. Blots were pre-blocked in PBS containing 5% (*w*/*v*) dried no fat milk and then incubated overnight at 4 °C with the specific anti-GluR1–4 antibody (see above). The anti-actin (Sigma) antibody was used as a control for loading. Antibody binding to blots were detected by chemiluminescence (GE Healthcare Life Sciences, Cologno Monzese, Italy).

### 2.7. Statistical Analysis

Statistical analyses were performed using the GraphPad PrismTM software (GraphPad Software, San Diego, CA, USA), as detailed in each paragraph.

## 3. Results

### 3.1. Sensitivity of Glioma Cell Lines to Perampanel, Analysis of Apoptosis, and Cell Cycle Perturbation

The cellular sensitivity of the human glioma cell lines U87, U138, A172, and SW1783 to perampanel was examined following different times of exposure (1 h and 24 h). A dose-dependent effect of perampanel was observed in all the tested glioma cell lines (Figure 1). The most marked anti-proliferative effect was detected when cells were exposed to perampanel for 24 h with small differences in IC_50_ values among the various cell lines (range: 178.5–258.5 µM). To determine whether the treatment resulted in an induction of apoptotic cell death, we performed a flow-cytometric analysis of apoptotic cells by PI/Annessin V assays. The results are summarized in Figure 2. Apoptosis was determined 24 h and 48 h after drug exposure start, i.e., after continuous 24 h exposure and 24 h after incubation in drug-free medium following the 24 h exposure. In all cell lines, the treatment with 250 µM perampanel produced a marked increase of apoptosis, with respect to the untreated cells. This increase was statistically significant in all cell lines when measured after 24 h of drug exposure; in A172 and SW1783, a statistically significant increase of apoptosis was detected even after 48 h or using a lower perampanel concentration, i.e., 100 µM (Figure 2).Low levels of necrosis were found in all cell lines (Appendix A), and, although perampanel could induce some necrosis, this effect was quantitatively marginal (always below 2%).

An analysis of cell cycle perturbations showed that, in U87 and U138 cell lines, although with some differences (250 µM at 48 h for U87 and 100 µM at 24 h for U138), treatment with perampanel resulted in a barely significant accumulation of cells in the G1 phase (Figure 3). However, no differences were detected in A172 or SW783 cell lines (data not shown).

### 3.2. Analysis of the Interaction between Perampanel and Temozolomide

Since temozolomide (an alkylating agent able to cross the blood-brain barrier) is widely used in the clinical treatment of GBM, we examined whether perampanel displayed a favorable interaction with temozolomide, using cell growth inhibition assays. U87, U138, and A172 cells were pre-treated for 1 h with perampanel and then exposed to increasing concentrations of temozolomide for an additional 72 h (Figure 4). Sensitivity to TMZ alone was in the high micromolar range, the A172 cell line being the most sensitive.

A synergistic interaction was detected in all cell lines, although to a different extent. The strongest effect was observed in A172 cells for the combination of 150 and 300 µM perampanel with temozolomide, with CI values ranging from 0.02 to 0.17 for the former concentration and from 0.04 to 0.33 for the latter. A very favorable drug interaction was also evident in U87 cells, in which all CI values except one (0.66) were lower than 0.5 (ranging from 0.16 to 0.43) for the combination with 150 µM perampanel. In U138 cells, only two CI values were indicative of marked synergism, i.e., 0.35 for the combination of temozolomide with 150 µM perampanel and 0.33 for the combination with 300 µM perampanel (Table 1). With the latter combination, in U138 cells, an antagonistic effect was also observed (Figure 4). Thus, pre-treatment with perampanel (even at the lower dose of 150 µM) significantly decreased the concentration of temozolomide needed to reach the IC_50_ (about 60- and 25-fold less for U87 and A172, respectively) (Figure 4, Table 1).

### 3.3. Effects of Perampanel on AMPA Receptors

Since AMPA receptors are heterotetrameric proteins consisting of tetramers of four different subunits (GluR1–4) [15], putative modulations in the expression of GluR1, GluR2/3, and GluR4 subunit were evaluated in U87 and U138 after cell exposure to perampanel. Treatment of U87 cells with perampanel resulted in an overall increased expression of all GluR subunits at both perampanel concentrations, as compared to control cells. The increase was particularly evident for GluR1 at 250 µM and GluR2/3 at 150 µM, as detected by flow cytometry after 24 h (Figure 5, panel a). Although with some discrepancies compared to flow cytometry, western blot analysis confirmed the trend toward increased GluR subunits expression, more evident for GluR1, after exposure of U87 to perampanel 150 µM (Figure 5, panel c, right).

A slight increase in GluR subunits after treatment with perampanel 250 µM was found in western blot analysis also in U138 (Figure 5, panel c left), although no differences in their expression was detected by flow cytometry in this cell line (Figure 5, panel b).

## 4. Discussion

After the 2009 clinical trial by Grossman [9], with median overall survival reaching 20 months in Talampanel-treated GBM patients vs. 14.6 months in the active treatment arm of the Stupp 2005 trial [16], and a proportion of 24-month survival of 40% vs. 27%, data from Iwamoto, showing no significant activity as a single agent in unselected recurrent malignant gliomas in terms of overall survival and progression free survival [17], appeared less promising, and no further clinical trials were performed with this molecule in the management of high grade glioma.

The development and availability of perampanel as an add-on treatment in epilepsy with focal onset has led to widespread clinical use of this drug, mainly for resistant and refractory epilepsies [18]. Although early data reported side effects related to mood alteration and aggressive behavior [19,20], more data on good tolerability and efficacy have now been collected [21,22], thus improving the safety of perampanel use.

In the present study, we used a preclinical pharmacology approach to examine the possible interest of perampanel in the treatment of human GBM, in which limited therapeutic options were available.

Our experiments have shown a potential antitumor activity of perampanel alone and a relevant synergistic effect of the combination of perampanel and temozolomide in high grade human glioma cells lines. This effect might be related to enhanced apoptosis of GBM cells, which are known for their marked resistance to apoptosis [23,24]. Perampanel was able to induce apoptosis in all the 4 glioma cell lines, although to different extents and with a more pronounced activity when higher drug concentrations (i.e., 250 µM) were used.

Our results are in line with those recently reported by Lange et al. [25], who investigated four low-passage GBM and three low-passage brain metastases lines and detected a cytotoxic effect by perampanel, but not by other investigated anti-epileptic drugs (levetiracetam, valproic acid, and carbamazepine). However, at variance with Lange’s results, we did detect an increase in apoptosis in perampanel-treated cells. Differences in the analyzed cell lines and detection of apoptosis by different methods, i.e., Annexin-V binding assay in our study and analysis of sub-G1 peaks in PI-stained cells in Lange’s work, may partly account for this discrepancy.

Although the concentration of perampanel used in our experiments is higher than the steady-state plasmatic concentration in patients with epilepsy (about 5 µM), several factors may create and maintain disequilibrium between the drug concentration in plasma and tissues, leading to drug concentration asymmetry [26]. Active uptake may be greater than efflux and other elimination mechanisms in the tissue, resulting in local concentrations that may be higher than those of plasma, as described in brain tissue for sunitinib and diazepam, for example [27,28].

Due to the role of glutamate receptors in proliferation and migration of glioma cells, we also analyzed whether perampanel had an effect on the expression of GluR subunits in two of the studied glioma cell lines. Under our experimental conditions, although with some differences between the two cell lines, western blot analysis showed that perampanel indeed upregulated the expression of the various GluR subunits in both U87 and U138 cells.

Indeed, modulation of AMPA receptor subunits has been described to modify the permeability of glioma cell to Ca++ and the overexpression of calcium impermeable AMPA receptors subunit, such as GluR2, inhibited glioma cell motility, and induced apoptosis [29]. In agreement with this mechanism, our results suggest that perampanel-induced modulation of GluRs subunits might decrease glioma cell permeability to calcium, resulting in induction of apoptotic cell death.

The finding of increased apoptosis after perampanel treatment of glioma cell lines, in addition to that of detection of a clear cytotoxic effect on the same lines and its longer half-life in humans compared to talampanel [11,12], makes perampanel require a more in depth investigation, with focus on its possible action from an oncological point of view in the management of GBM patients with epilepsy.

The major advance in therapy of GBM in the last 20 years has derived from the addition of temozolomide, an alkylating agent, to the standard of care [16] that, until 2005, was represented by maximal safe surgery, followed by radiation therapy; however, survival has not increased meaningfully. Therefore, drugs able to enhance the cytotoxic activity of temozolomide are of obvious interest and the results presented here indicate perampanel as a promising molecule.

Several studies have investigated the topic of antiepileptic drugs in the context of glioma management, trying to unravel whether some of these drugs might exert an anti-neoplastic activity in addition to the antiseizure action; in this respect, valproate has been the focus of major interest, due to its activity as a histone-deacetylase modulator [30]. However, pooled analysis of clinical data has failed to prove a clear-cut survival benefit in those GBM patients receiving valproate on top of surgery, temozolomide, and radiation therapy, stressing the need for further randomized trials on this topic [31].

The more recent introduction of perampanel in the armamentarium of antiepileptic drugs used in the general epilepsy population has, so far, precluded an analysis of its possible effects on oncological outcomes in glioma, with available studies focusing on anti-epileptic efficacy and tolerability in patients with brain tumors [32,33].

When considering the potential combined anti-epileptic and anti-tumor activity of perampanel in glioma, it might be speculated that IDH1-mutated low-grade gliomas may benefit most in terms of reduction in seizure activity [34]. IDH1 status was known only in U87 cell line (wild type); in high-grade gliomas, Buckingham has shown high levels of glutamate release by tumor cells in an experimental model of brain tumor-associated epilepsy [35]. Reduction of glutamate release in this model has significantly reduced seizure frequency. The same group has reported reduced proliferation in U87 by the glutamate transporter inhibitor sulfasalazine [36].

Our preliminary data, as well as those reported from Lange, make it worth investigating, especially in the clinical scenario, a putative effect of perampanel on glioma progression and consequently on progression free survival and overall survival.

## Figures and Tables

**Figure 1 jpm-11-00390-f001:**
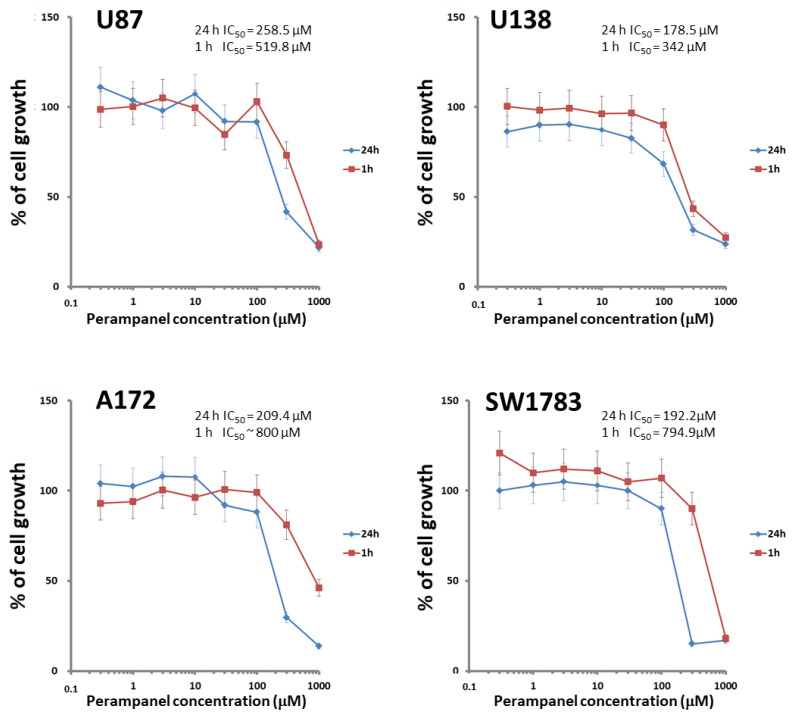
Sensitivity of glioblastoma cell lines to perampanel. Cell sensitivity to the drug was assessed by cell growth inhibition assays after exposure of cells to increasing perampanel concentrations for 1 h (red line) or 24 h. (blue line). The IC_50_ value (i.e., concentration of perampanel inhibiting cell growth by 50%) is reported. The results are expressed as percent of growth inhibition. The results are from at least three independent experiments. Error bars represent the SD values.

**Figure 2 jpm-11-00390-f002:**
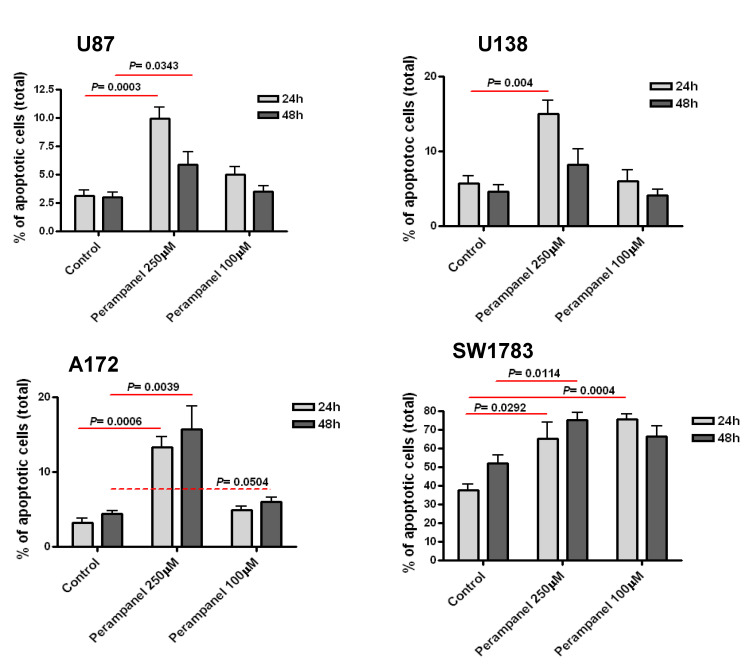
Analysis of perampanel-induced apoptosis in glioblastoma cell lines. U87, U138, A172, and SW1783 glioma cell lines were cultured in complete medium containing perampanel 250 µM and 100 µM, approximately corresponding to IC_50_ and IC_20_, respectively. Cells were harvested after 24 h (light gray) and 48 h (dark gray), and apoptosis was measured by Annexin V-binding assay. The results are expressed as a percent of apoptotic cells. P values were calculated using the two-sided Student’s *t* test. The reported values refer to at least three independent experiments. Error bars represent the SE values.

**Figure 3 jpm-11-00390-f003:**
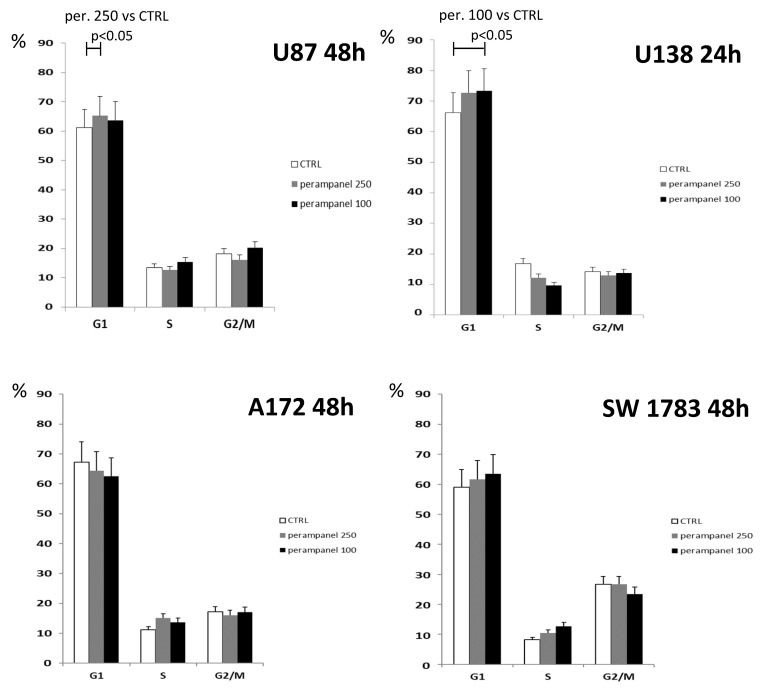
Cell cycle distribution of perampanel-exposed glioblastoma cells. Unsynchronized cells were cultured in complete medium alone (empty bars), and 100 µM (black bars) or 250µM perampanel (grey bars) was added for 24 h. The drug was then removed, and fresh medium was added. The cells were harvested and analyzed for cell cycle distribution 24. h or 48 h later. The results are expressed as the percent of cells in the specific phase of the cell cycle (G1, S, or G2/M). Statistically significant differences were detected using the two-sided Student’s *t* test. The reported results refer to at least three independent experiments. Error bars represent the SD values.

**Figure 4 jpm-11-00390-f004:**
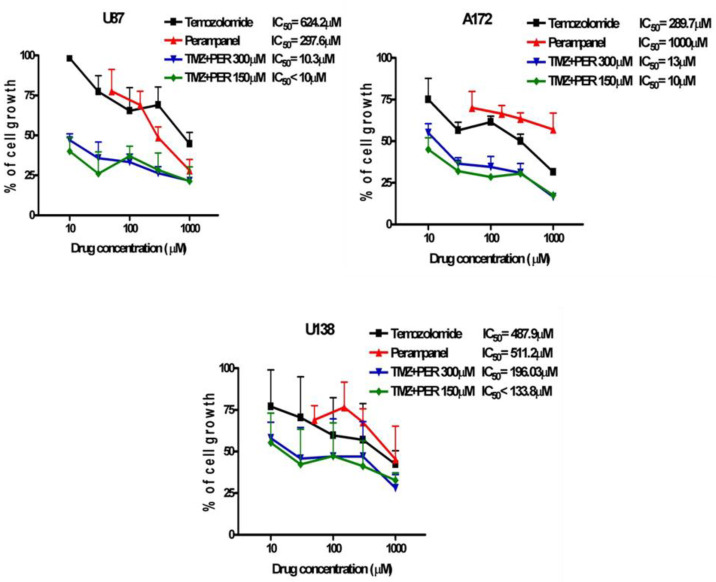
Synergistic effect of the combination of perampanel and temozolomide. Glioblastoma cell lines were pre-treated for 1 h, with perampanel (PER) at 150 µM or 300 µM, and then cultured in medium containing an increasing concentration of temozolomide (TMZ).

**Figure 5 jpm-11-00390-f005:**
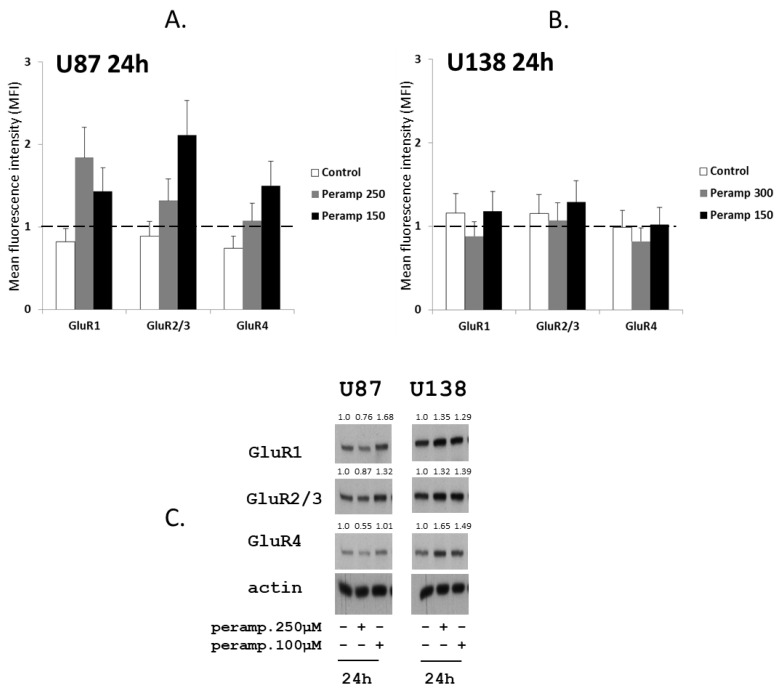
Perampanel modulates the expression of glutamate receptors: (**A**,**B**) Cell lines were cultured for 24 h in complete medium (white histograms) and in medium containing 250 µM (gray histograms) or 150 µM perampanel (black histograms). Cells were then harvested and analyzed by flow cytometry using receptor subunit-specific antibodies. Data in panel a and b are expressed as the ratio between the mean fluorescence intensity (MFI) of the specific antibody and the MFI of the relative isotypic control (see methods). (**C**) Western blot analysis of GluRs was carried out in untreated cells and cells exposed to perampanel, as indicated. Control loading is shown by actin. The protein band intensity was quantified using ImageJ, normalized to that of the respective loading control and expressed relative to the level of control cells (set to 1).

**Table 1 jpm-11-00390-t001:** Combination index. After 72 h of drugs exposure, cells were harvested and counted and the combination index was calculated according to the Chou–Talalay method using the Calcusyn software (Biosoft, Cambridge, UK), which assigns a combination index value (CI) to a drug combination. A CI value lower than 0.85–0.90 indicates synergistic drug interactions, whereas CI values higher than 1.20–1.45 and around 1 stand for antagonism and additive effect, respectively. The reported values refer to at least three independent experiments. Error bars represent the SD values.

Combination Index						
	U87		A172		U138	
	per 300 μM	per 150 μM	per 300 μM	per 150 μM	per 300 μM	per 150 μM
**TMZ 1000 μM**	0.63 ± 0.4	0.66 ± 0.6	0.06 ± 0.04	0.09 ± 0.1	0.33 ± 0.3	0.82 ± 0.7
**TMZ 300 μM**	0.44 ± 0.1	0.43 ± 0.3	0.21 ± 0.2	0.17 ± 0.09	2.34 ± 3.6	0.56 ± 0.6
**TMZ 100 μM**	0.49 ± 0.1	0.35 ± 0.1	0.11 ± 0.05	0.04 ± 0.03	2.22 ± 3.7	0.64 ± 0.9
**TMZ 30 μM**	0.48 ± 0.2	0.16 ± 0.1	0.04 ± 0.03	0.02 ± 0.02	1.14 ± 1.9	0.35 ± 0.6
**TMZ 10 μM**	0.86 ± 0.1	0.30 ± 0.1	0.33 ± 0.1	0.07 ± 0.08	1.39 ± 1.4	0.94 ± 1.5

## Data Availability

All data generated or analyzed during this study are included in this published article. Raw and processed data are stored in the laboratory of the corresponding authors and are available upon request.

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
