# Peer review of "Synergistic Effect of Perampanel and Temozolomide in Human Glioma Cell Lines"

_jpm, 2021, doi:10.3390/jpm11050390_

Round 1

Reviewer 1 Report

In the manuscript titled "Synergistic effect of perampanel and temozolomide in human glioma cell lines" prepared by Salmaggi et al., the author tested the therapeutic efficacy of perampanel and temozolomide in glioma cell lines. They discovered a strong synergistic effect of this combination in U87 and A172, but not U138 cells. They propose that the therapeutic effect is through increased GluR2/3 expression. I have several concerns as follow:

  1. The author conducted a series of dose-response studies to show the pharmacodynamics of therapeutic compounds. However, the author needs to provide additional evidence showing the synergistic effect between temozolomide and perampanel, such as combination index or dose matrix.
  2. The concentration of perampanel used in the present study is rather high (> 100 uM). Whereas this drug is used in a relatively small dose in clinical practice (8-12 mg/day). The author may want to discuss whether the dosage issue could affect future clinical practice.
  3. The author claimed GluR2/3 is involved in the cytotoxicity of perampanel. Has the author performed a loss-of-function study such as small interference RNA?
  4. Several recent studies revealed the heterogeneity of glioma cell lines to temozolomide, especially for those with IDH mutation background (PMID: 20975057, PMID: 28202508). As some of these tumors are prone to epileptic activity (PMID: 28404805, PMID: 32825279), the author may want to discuss the relevance of glioma subtypes in the context of using perampanel. 

Author Response

Answer to Reviewer 1

  • The author conducted a series of dose-response studies to show the pharmacodynamics of therapeutic compounds. However, the author needs to provide additional evidence showing the synergistic effect between temozolomide and perampanel, such as combination index or dose matrix.

We thank the reviewer for the insightful comments. Regarding the synergistic effect we have elected to summarize the results according to the recommendation of the following paper: Bijnsdorp, I. V., Giovannetti, E., & Peters, G. J. (2011). Analysis of drug interactions. Methods in molecular biology (Clifton, N.J.), 731, 421–434. https://doi.org/10.1007/978-1-61779-080-5_34. Here the suggestion is to provide the mean value of CI at the tested combinations. Thus, the combination index values have been presented in Table 1.

  • The concentration of perampanel used in the present study is rather high (> 100 uM). Whereas this drug is used in a relatively small dose in clinical practice (8-12 mg/day). The author may want to discuss whether the dosage issue could affect future clinical practice.

The concentration of perampanel used for this study is indeed 10 to 20-fold higher than the steady-state plasmatic concentration in patients with epilepsy (about 5 µM) and previous data obtained in normal brain tissue in rats, reported a brain-to-plasma ratio of 0.62 for peramapanel (Paul D et al., J Pharm Biomed Anal . 2018). However, several factors may create and maintain disequilibrium between the drug concentration in plasma and tissue, leading to drug concentration asymmetry (Zhang et al., Drug Metabolism and Disposition October 2019). Predicting tumor tissue drug distribution is challenging especially in glioblastoma in which the blood brain barrier is often altered. Active uptake may be greater than efflux and other elimination mechanisms in the tissue resulting in local concentrations that may be higher than those of plasma as described in brain tissue for sunitinib and diazepam for example (Chee EL, Eur J Drug Metab Pharmacokinet, 2016; Geoffrey J et al. Neuroreport. 2010).

We added a short paragraph in discussion (5th paragraph, page 10).

  • The author claimed GluR2/3 is involved in the cytotoxicity of perampanel. Has the author performed a loss-of-function study such as small interference RNA?

The Reviewer raises an excellent point. Here, we elected to use a pharmacological approach instead of a genetic approach in an attempt to generate results translatable to the clinical context. We avoided to knock down the GRM2 and GRM3 genes given that studies in knock out mice have provided evidence of behaviour impairment (see https://doi.org/10.1038/npp.2011.145 and https://doi:10.1186/1756-6606-7-31)

  • Several recent studies revealed the heterogeneity of glioma cell lines to temozolomide, especially for those with IDH mutation background (PMID: 20975057, PMID:28202508). As some of these tumors are prone to epileptic activity (PMID: 28404805, PMID: 32825279), the author may want to discuss the relevance of glioma subtypes in the context of using perampanel.

The topic has been addressed in the discussion (page 11)

Reviewer 2 Report

Thank you for the opportunity to review the manuscript. The authors of the manuscript "Synergistic effect of perampanel and temozolomide in human glioma cell lines" presented a very interesting approach to the treatment of glioblastoma involving the addition of perampanel to temozolomide (TMZ) treatment. The results obtained are interesting and could certainly be of interest to many scientists and oncologists working in this field. 

Unfortunately, the study is not well planned and presented, the main complaints are as follows:

  • line U138 was reported as TMZ resistance cells. therefore why did the authors choose it for this study?
  •  SW1783 is not glioma cell line, is astrocytoma (grade III)

 I therefore recommend the following major revisions:

  • change cell lines or justify their selection
  • describe the sensitivity / resistance of the tested cell lines to TMZ
  • compare the obtained IC50 TMZ results in the tested lines to the literature data (the results do not coincide, especially for the U87 line, why?)
  • what kind of assay was used to evaluate cell growth (MTT, etc.). this should be described in the methodology.
  • Figure 2 - on the y axis the maximum values should be the same. Moreover, the results are incomplete. Essential data on necrotic and living cells are missing.
  • Figure 3 - Statistically significant differences are marked in an unreadable way.
  • Figure 4 - on the y axis the maximum values should be the same. 
  • why is the SW1783 line not used in every assay?
  • Figure 5 a and 5b - what unit is on the y axis?
  • the discussion is insufficient and leaves us unsatisfied. there is no interpretation and explanation of the obtained results, emphasising the differences between the cell lines tested.

Author Response

Answers to Reviewer 2

  • change cell lines or justify their selection

As suggested we specified the origin of SW1783 and briefly explained the choice (last paragraph of the introduction and in M&M).

Since the major focus of this work was investigation of a putative anti-tumor effect of perampanel in vitro, we chose to study only high-grade glioma cell lines.

  • describe the sensitivity / resistance of the tested cell lines to TMZ

As suggested we have included a description of sensitivity to TMZ alone (see page 6, end of first paragraph)

  • compare the obtained IC50 TMZ results in the tested lines to the literature data (the results do not coincide, especially for the U87 line, why?)

The reviewer raises a point that is as interesting as it is difficult to argue. Several factors may influence the in vitro cell sensitivity to drugs including cell line passage number, time of drug exposition, culture medium and serum composition. In our experimental conditions, TMZ cytotoxicity (IC50) was measured after 72h drug exposition resulting in a median IC50 of 624 µM for U87 and 489µM for U138 (see fig. 4).

Previously published results obtained in our lab on these cell lines showed a similar TMZ IC50 for U138 (about 650µM) and quite a different IC50 for U87 (about 200µM) (Balzarotti M et al., Oncology Res 2004). Accordingly, Ryu CH et al, using similar experimental conditions, reported a TMZ IC50 of about 250 µM in U87 and about 750µM in U138. Our data seems approximately in line for U138. For U87 the reviewer is right, we too noticed the discrepancy which however was reproducible in various experiments. Planning this study and having the financial support, we decided not to use frozen aliquot but to get new bulk from ATCC of each cell lines involved and use them within 20 passages as stated in M&M. This may perhaps explain the differences from our previously published data but it is still difficult to discuss,

  • what kind of assay was used to evaluate cell growth (MTT, etc.). this should be described in the methodology.

The assay used to evaluate the effect of the drugs is a cell growth inhibition assay based on cell counting with an automatic instrument (see paragraph 2.2)

  • Figure 2 - on the y axis the maximum values should be the same. Moreover, the results are incomplete. Essential data on necrotic and living cells are missing.

As suggested, figure 2 has been modified accordingly. Data regarding necrotic and live cells have been summarized in a supplementary figure not for making figure 2 difficult to read

  • Figure 3 - Statistically significant differences are marked in an unreadable way.

Figure 3 has been modified accordingly

  • Figure 4 - on the y axis the maximum values should be the same. 

Figure 4 has been modified accordingly

  • why is the SW1783 line not used in every assay?

Given that the SW1783 cell line was already quite sensitive to perampanel induced apoptosis it was not included in drug combination assays

  • Figure 5 a and 5b - what unit is on the y axis?

We apologise for the inaccuracy. Figure 5 has been edited accordingly.

  • the discussion is insufficient and leaves us unsatisfied. there is no interpretation and explanation of the obtained results, emphasising the differences between the cell lines tested

The discussion has been expanded with as emphasis on anti-seizure and anti-tumor activities of perampanel as a putative modulator of the glutamate pathway. Discussing further on differences among lines has the risk to be too speculative and not supported by the available data.

Round 2

Reviewer 1 Report

The author made improvements to the manuscript.

Reviewer 2 Report

The manuscript may be published